# Barriers Involved in the Safety Management Systems: A Systematic Review of Literature

**DOI:** 10.3390/ijerph19159512

**Published:** 2022-08-03

**Authors:** Weiliang Qiao, Enze Huang, Hongtongyang Guo, Yang Liu, Xiaoxue Ma

**Affiliations:** 1Marine Engineering College, Dalian Maritime University, Dalian 116026, China; xiaoqiao_fang@dlmu.edu.cn (W.Q.); ghty@dlmu.edu.cn (H.G.); 2School of Maritime Economics and Management, Dalian Maritime University, Dalian 116026, China; huangenze@dlmu.edu.cn (E.H.); liuyang0120@dlmu.edu.cn (Y.L.); 3Public Administration and Humanities College, Dalian Maritime University, Dalian 116026, China

**Keywords:** safety barrier, barrier management, risk propagation, bibliometrics, research perspective

## Abstract

Safety barriers are widely accepted in various industries as effective risk management tools to prevent hazardous events and mitigate the consequences caused by these events. Studies on safety barriers have been increasing in recent decades; therefore, the general idea of this article is to present a systematic review of the field. The purpose of this article is threefold: (1) to map various networks for the barrier-related articles collected from WoS; (2) to summarize the advances of the safety barrier at both the individual level and barrier management level on the basis of six issues, and (3) to propose the research perspectives associated with safety barriers considering the latest theories and methodologies in the field of safety management. Based on the findings and insights obtained from the literature collected by a bibliometric and systematic review, studies on barrier management within the complex socio-technical system are analyzed, and the framework of “risk-barrier capacity” is proposed for future development, in which the challenges stemming from industrial intelligence may be solved through resilience theory. Meanwhile, intelligent technologies are also able to serve as health status monitoring devices for various barrier elements.

## 1. Introduction

Over the past decades, a growing interest in risk analysis and system reliability has been observed in various industries. The risks are not accepted to some extent within the framework of safety engineering, in which hazards are strictly mitigated. As a result, inherently safe design is popular to mitigate the potential risks involved in operational actions [1], especially in the case of various advanced technologies applied in practical engineering. However, the introduction of advanced technologies has not substantially improved the safety level, which has been argued by Utne et al. [2] and Ventikos et al. [3] for risk-associated issues with the objective of developing autonomous vessels. The reason is generally accepted as the twinning of uncertainty and complexity, which makes incidents nearly inevitable [4]. For instance, according to the statistics by HIS for maritime shipping, for accidents that occurred from 2010 to 2020, as illustrated in Figure 1, the number of accidents is still not satisfactory even though various advanced technologies and regulations/codes/conventions at the national and international level have been implemented. The aforementioned uncertainty and complexity are generally considered typical characteristics of complex socio-technical systems in which the interaction of humans, the environment and machinery should be emphasized [5]. However, the solution for uncertainty and complexity cannot be accomplished in one stroke, and the balance between cost and effectiveness has to be considered [6,7]. Based on the review by Puisa et al. [8], safety strategies coping with uncertainty and complexity can be classified as preventive and mitigative; the former is focused on the prevention of incidents or hazards, and the latter is concentrated on alleviating the consequences of accidents or hazardous events, both of which exactly correspond to the functions of preventive safety barriers and protective safety barriers, respectively. Safety barriers include physical or nonphysical approaches to defend against external or internal hazards [9]. The investigation of interaction and coordination among various safety barriers is essentially attributed to a kind of issue that can be solved from the perspective of complex socio-technical systems due to numerous heterogeneous safety barriers being contained, such as human-related, organizational and technical barriers [10]. In addition, studies on safety barriers could generally transfer academic attention from hazards or risks that are negative to positive aspects of the functional system; as a result, a great deal of data associated with safety barriers can be collected to support decision-making.

The wide application of safety barriers in various industries can be found in the existing literature. As early as the 1960s [11,12], the concept of accidents as abnormal or unexpected states of a system was developed, which contributed to the first definition of safety barriers made by Taylor [13] as “equipment, constructions, or rules that can stop the development of an accident”. Later, the principle of the Swiss cheese model [14] was considered in the studies of safety barriers, which can be seen as cheese slices in this model, and various classifications and definitions for safety barriers were subsequently proposed, which can be found in Section 3.1.1. Based on the general classification of safety barriers as preventive and protective, the Bow-tie diagram is combined with safety barriers [15]; as a result, the concept of barrier management is first initiated and applied in the rig industry [16]. The functions of safety barriers in preventing the possibility of issues and attenuating the consequences of accidents have been verified in various industries, such as offshore oil and gas [17], fire prevention [18], nuclear power [19], maritime shipping [20], Natech (natural-hazard-triggered technological) events [21] and road transportation [22]. According to Puisa et al. [8], as illustrated in Figure 2, different kinds of safety barriers are applicable for specific industrial scenarios; for instance, in the offshore oil and gas industry, mitigative safety barriers are more popular, while preventive safety barriers are preferred in the railway industry. In practice, a project named the accidental risk assessment methodology for industries (ARAMIS) was launched in January 2002 to develop a new risk assessment methodology based on deterministic risk-based approaches [23], following which the PSA [24] released a regulation proposing six steps to implementing safety barriers. A corresponding barrier management solution for minimizing risks and preventing incidents in the oil and gas industry was developed by DNV GL [25]. As discussed, safety barriers have been widely studied and applied in various industries. However, the great potential of the safety barriers in the field of safety management needs to be further released, especially in the era of Industry 4.0. For this purpose, it is necessary to implement a systematic review of the advances in safety barriers, based on which future research on safety barriers can be summarized. Finally, the results of this literature review are expected to improve safety management in Industry 4.0.

This paper is mainly aimed at addressing the issues associated with safety barriers in the areas of definition, classification, research topics, advances, and future perspective studies. For this purpose, CiteSpace is applied in this study to implement bibliometric analysis and research mapping. There have already been several review articles on the topic of safety barriers; for example, Sklet [9] performed a comprehensive review of the aspects of the definition, classification and performance of safety barriers. Later, the performance assessment of fire protection-related safety barriers was summarized by Gomez et al. [26], and most recently, Liu [27] reviewed the advances associated with safety barriers for the topics of theory, engineering and management. The present study has the following distinct features:

(1) This study is the first to apply bibliometrics to map academic networks for the topic of safety barriers and barrier management in terms of authors, affiliations, citations, co-occurrences and co-citations.

(2) The issues of barrier management involved in industrial practice are identified on the basis of the proposed safety barrier management system.

(3) The research perspectives in this paper bridge the gaps between academic research and industry application by the proposed safety management framework of “risk-barrier capacity”.

The remainder of this paper is organized as follows: bibliometrics and research mapping for safety barriers will be discussed in Section 2, and the advances focused on safety barriers are then summarized in Section 3. In Section 4, research perspectives on safety barriers are proposed, and finally, the conclusions are presented in Section 5.

## 2. Bibliometrics and Research Mapping on Issues of Safety Barriers

### 2.1. Dataset Preparation

Bibliometrics are frequently applied to quantitatively describe bibliographic information on the aspects of scientific production, citations, affiliations or keywords that can be presented in the form of visualization, such as maps, graphs and networks [28,29]. Generally, dataset preparation is the precondition for the implementation of bibliographic analysis. In the present study, the *Web of Science* is selected as the source of the dataset collection because most high-quality and important scientific productions are indexed in this database [30]. The setup for the collection is presented in Table 1, in which the core collection refers to the *science citation index expanded (SCI-E), social sciences citation index (SSCI)*, *arts and humanities citation index (AHCI)* and *emerging sources citation index (ESCI)*. The keyword “barrier management” is also employed during dataset collection based on the widely accepted fact that the interaction and coordination among various safety barriers must be considered [31]. After searching the WoS according to the protocol shown in Table 1, the initial database for this study was developed with 183 research articles. By reviewing these articles, it can be found that some articles have little or no correlation with safety management or risk mitigation, for instance, the barriers have the meaning of “obstacle” in some articles. Therefore, it is necessary to filter these articles having little or no correlation with the research objectives of this study by focusing on the titles, abstracts and keywords. In detail, whether a research article is included in the analysis is determined by the following: (1) articles in the fields of public health or disease prevention are eliminated; (2) studies associated with anthropomorphic dummies are excluded from the initial database and (3) research articles in the field of biological sciences, material sciences and chemistry are not within the scope of this study. Therefore, the contents of the initial database are refined from 183 documents to 113, which are regarded as the final dataset and forwarded for further analysis.

To obtain the general distribution of safety barrier research, all the filtered documents are browsed and assigned to suitable application scenarios that are determined by the studied objects. The results are shown in Figure 3. The scientific contributions associated with safety barriers are largely in the field of offshore oil and gas and transportation (including roads, shipping, air and rail), where published papers account for approximately 62% of the total scientific production. To clarify the distinctions between categories with potential overlaps, classification principles may be proposed as follows: (1) the studies associated with the prevention of various accidents, including road injuries, marine disasters and firefighting, are classified into accident prevention, such as the research work by Bellamy et al. [32] and Lenoble et al. [33]; (2) the papers that focus on the principle and assessment of safety barriers are categorized as general, in which review articles are also included; (3) the papers on issues that appeared in oil and gas processing in land-based facilities are attributed to the category of the chemical industry, while the production of oil and gas at sea is considered in the offshore oil and gas group; for instance, the study associated with hydrogen application implemented by Duijm and Markert [34] is grouped as the chemical industry.

### 2.2. Distribution of Authors and Affiliations

In the investigated period, 328 authors contributed to studies on safety barriers, and the co-authorship is depicted in Figure 4, which is mapped on the basis of scholar collaborations for publications with the threshold setup of three. There are a total of eight clusters illustrated in Figure 4, and it is interesting to note that the largest cluster is represented by Gabriele Landucci and Valerio Cozzani, who are also the two most productive scholars in Italy. The cluster represented by Genserik Reniers and Nicola Paltrinieri is ranked as the second-largest cluster, which is linked with the largest cluster associated with the co-authorship between Gabriele Landucci and Nicola Paltrinieri. Some scholars suspended their studies on the topics of safety barriers. Another remarkable phenomenon observed in Figure 4 is that co-authorship can hardly be observed between productive authors even though they share similar research interests, such as Faisal Khan and Marvin Rausand, both of whom concentrate on safety barriers for deepwater drilling operations. Overall, the clusters for co-authorship are characterized by agglomeration effects for different industries.

In most cases, the academic contribution of authors is closely related to the countries and institutions to which they affiliate. The abovementioned 328 authors belong to 159 institutions in 39 different countries. In the present study, institutions with more than four papers are selected as the analysis sample, which contains 13 institutions and 10 countries with 92 papers accounting for 81.4% of the total publications. The distribution of academic production by institutions and countries is illustrated in Figure 5. It should be noted that China is absent from the productive countries presented in Figure 5 even though China is ranked as the fourth most productive country, with 12 papers published. The reason for this is that there is no institution or scholars in China regarded as productive in the field of safety barriers. A similar situation applies to the case of Brazil. It is interesting to find that all the academic production of Norway (the second most productive country) is mainly attributed to NTNU, which is regarded as the most productive institution globally, with 16 publications. The situation for the Netherlands is more interesting in that all the academic contributions associated with safety barriers are made by the *Delft University of Technology*, which is famous for advanced engineering technology. In addition, Italy is observed as being the most productive country where the publications mainly come from the *University of Bologna*, the *University of Pisa*, and *Polytechnic University of Milan*. According to the statistics illustrated in Figure 5, all the institutions, with the exception of Memorial *University Newfoundland* (Canada) and *Queensland University of Technology* (Australia) in the analysis sample, are scattered across Europe, accounting for 89.1% of the publications contained in the sample.

### 2.3. Scientific Production and Citations

According to the publication year of the collected articles and their citations obtained from the *Web of Science*, Figure 6 can be obtained. Even though the concept of safety barriers was proposed as early as 1988 [13], until 2002, when a European project named accidental risk assessment methodology for industries (ARAMIS) was launched by 15 partners from 10 European countries [23], the application potential of safety barriers was explored systematically, and three years later, the ARAMIS project was ready to be applied in the offshore oil and gas industry. From then on, as illustrated in Figure 6, the number of publications associated with safety barriers is observed to have an upwards trend, and an increasing number of scholars pay attention to the definition, principles and effectiveness of safety barriers, which is verified by the trend of statistical citations presented in Figure 6. The upwards trend of citations is almost continuous from 2006 to 2020. Scientific production in the fields of safety barriers can be generally divided into two stages, with 2010 being the demarcation point: the period before 2010 is regarded as the first period when publications were observed to be decreasing from 11 papers in 2006 to two papers in 2010. In this stage, inspired by the practice of ARAMIS, the application of safety barriers was explored preliminarily in various fields, such as offshore oil and gas [35], roads [36] and chemical industries [34]. Then, a visible increase in scientific production was observed starting in 2011 compared to the previous year, which increased by 300%. The upwards trend that initiated in 2011 until today is probably due to the benefits of safety barriers being verified by pioneering works implemented academically and practically in the first stage. The frequency of citations illustrated in Figure 6 can also be used to verify the importance of safety barriers in various industries. In addition, the citations of the papers contained in the database sample are analyzed by establishing the network involving these papers citing each other, and there is a total of 63 papers being mapped due to the existing connection between them. The results are presented in Figure 7. The top 10 cited papers on the issue of safety barriers are extracted from the database sample and presented in Table 2.

The most globally cited paper is “*Safety barriers: Definition, classification, and performance*”, which is a review conducted by Sklet [9] with 190 citations (11.88 per year on average). Notably, Dianous and Fievez [37] systematically explained the principle of the ARAMIS project, with their work cited 126 times in the analyzed period, the second most cited paper. In addition, the paper titled “*Barrier and operational risk analysis of hydrocarbon releases (BORA-Release) Part I. Method description*” [38] is also frequently cited by scholars due to its good explanation of the principle of safety barriers defending against operational risks. According to the mapping of citations illustrated in Figure 7, Snorre Sklet and Gabriele Landucci are the two most cited scholars in the citation network. The contribution of Snorre Sklet from NTNU is largely due to his review article titled *Safety barriers: Definition, classification, and performance*, which is regarded as the most cited paper, and active collaborations with scholars in Northern Europe can be observed, such as with Aven Terje from the University of Stavanger and Nijs Jan Duijm from the *Technical University of Demark*, all of whom are known due to their contribution to the field of risk analysis and reliability. Therefore, the scientific contribution made by these scholars in Northern Europe can be regarded as an obvious cluster within the network. Another cluster is developed with the papers associated with Landucci [18], and Gabriele Landucci (affiliated with *Università di Pisa*, Italy), who are also ranked among the most productive scholars.

### 2.4. Sources Analysis

The papers contained in the developed database in this study are attributed to 47 different journals and published in the 16-year period from 2005 to 2021. As one of the important elements of bibliometric analysis, the sources of the papers involved in the database sample are analyzed; the top 10 journals with more than three articles are presented in Figure 8. These top journals are widely regarded as being of high quality in the field of risk analysis and safety, including the *Journal of Loss Prevention in the Process Industries, Safety Science, Reliability Engineering and System Safety*, and *Accident Analysis and Prevention*. Meanwhile, the network of paper sources based on citations is mapped, as shown in Figure 9.

According to Figure 8, the most relevant source is dominated by the *Journal of Loss Prevention in the Process Industries* with 16 publications, which is followed by the journals *Reliability Engineering and System Safety* and *Safety Science* with 12 papers. The total number of papers published in the aforementioned three journals accounts for approximately 39% of all papers in the database sample. In addition to the journals listed in Figure 9, there are still 30 other journals with only one article published. Even though Reliability and System Safety is not ranked first, it is regarded as the most comprehensive journal, involving seven different fields, with only the field of offshore oil and gas missing. However, approximately 45% of all papers on issues in the field of offshore oil and gas are published in the *Journal of Loss Prevention in the Process Industries*, in which there are no articles in the fields of maritime shipping and accident prevention. In addition, the articles published in Safety Science are mainly focused on the topics of review, assessment and the leading descriptions of safety barriers, which are attributed to the general group. Furthermore, it is interesting to observe that there are journals that only publish scientific research from industrial fields; for instance, all the papers published in *the International Journal of Crashworthiness, Engineering Failure Analysis*, and *Accident Analysis and Prevention* are associated with road transportation. The articles’ sources can be further analyzed by mapping the network with regard to their co-citations, in which the relation between two different publications can be evaluated according to the number of documents citing both of these papers [44]. As a result, the mutual relationships between paper sources are visualized in Figure 9. A total of 30 sources are selected in this study according to the citation ranking for the different sources, and the weights are represented by the value of citations.

According to Figure 9, the most relevant sources are *Reliability Engineering and System Safety* and *Journal of Hazardous Materials*, and generally, the well-known safety-related journals are closely connected based on co-citations, except for the *International Journal of Crashworthines,* which mainly published articles associated with road transportation. It is noted that the cluster centered on *Reliability Engineering and System Safety* is the largest cluster, even though the number of articles published is ranked third. A similar situation can also be found in *Accident Analysis and Prevention* and *Risk Analysis*, both of which are characterized by noticeable co-citations with few articles published. In addition, as the mainstream source for topics of marine engineering, *Ocean Engineering* has recently been presented in clusters connected with *Safety Science* and the *Journal of Loss Prevention in the Process Industries*.

### 2.5. Thematic Analysis

The keywords, including the author’s keywords and keywords plus, are statistically used to illustrate the frequency of words used by scholars. Based on the number of mutual occurrences in the articles contained in the database, the co-occurrences of these keywords are mapped in Figure 10, in which the width of connection indicates the number of mutual co-occurrences, and the frequency of a single keyword appearing in the database is represented by the radius of the circle. In addition, the color of the connections is used to imply the time when the mutual co-occurrences of the two keywords occurred. To better understand the research tendency of safety barriers from the perspective of keywords, the time is set up as the analysis axis, and the results of reorganizing the keywords are presented in Figure 11.

According to the contents involved in Figure 10, the majority of terms in this study are associated with issues of safety and risk. As mentioned in Section 2.1, “safety barrier” and “barrier management” are selected as the keywords to search the *WoS*; however, the frequency of “barrier management” is much lower than that of “safety barrier”. It is interesting to discover that the appearance of “accidents” in this network is not noticeable, which indicates that safety barrier studies have emphatically improved the system in terms of safety and reliability with the recognition that accidents are undesired conditions of the system. Based on the location of “accident” in the network illustrated in Figure 10, the function of “accident” can be considered to verify or analyze the performance of safety barriers because of its connection between “safety barrier” and “performance”. In addition, the keyword “*model*” is critical for the topological characterization of the network illustrated in Figure 10, and the modeling of “safety barriers” can be applied to develop barrier management systems, evaluate barrier performance, and establish safety strategies. The terms *domino effect, cascading events* and *quantitative risk assessment* indicate the recent application of safety barriers to interrupt risk propagation in domino events, and the presence of Bow-ties and diagrams implies that the study of safety barriers is frequently mapped into the Bow-tie diagram to identify the functions of safety barriers and implement risk assessment and analysis. In Figure 11, the keywords are generally clustered into 4 stages based on time. Based on the keywords that appeared in the first stage from 2005 to approximately 2008, it can be inferred that the concentration of barrier studies is focused on the chemical industry, such as *hydrocarbon release*. Then, with the development of a general theory for safety barriers, the concept of barrier management proposed in 2015 signified the second period of safety barrier research during which *organizational factors* known as nontechnical factors are fully considered for *risk management*. Regarding the third stage, in 2018, research on safety barriers became extensive, and various models were developed to cope with the performance assessment and risk analysis in different scenarios, such as domino events [45], offshore oil and gas [46], and the chemical industry [47]; however, advanced techniques aimed at addressing safety barriers are still undeveloped even though some methodologies involving Bow-tie diagrams and probabilistic-based methods have been explored to implement quantitative risk assessments, such as [10,48,49]. In the most recent stage, starting in 2020, the dynamic Bayesian approach is applied to the issues of safety barriers, which can be considered the beginning of advanced technologies being applied in the field of safety barriers. In the near future, it can be reasonably inferred that there will be an increasing number of advanced technologies utilized to quantitatively analyze the issues associated with safety barriers.

## 3. Advances Focused on Safety Barriers

According to the review by Sklet [9], the function of safety barriers is mainly to prevent, control or mitigate undesired events or accidents, which can be considered preventive aspects of safety barrier functions. The function of safety barriers can essentially also be represented by the protective aspect of attenuating the adverse effects stemming from unexpected events or accidents [10]. Based on the collected articles in this study, safety barrier-related works can be grouped at the individual level and management or system level, as a result, the advances in safety barriers can be discussed from two topics: barriers at the individual level and barrier management level. As discussed above, a safety barrier was initially proposed to defend against undesired events or risks; therefore, studies associated with safety barriers at an early stage are focused on the functioning of individual barriers or technical barriers, such as [9,36,42]. Later, the concept of a complex socio-technical system appeared in the field of risk and reliability, where single or technical barriers can hardly cope with hazardous events that occur in industrial operation, especially the risks associated with human factors [50]. Therefore, different groups of safety barriers must be integrated to mitigate undesired events, and a group of safety barriers can be regarded as barrier systems that are designed and implemented to perform multiple safety barrier functions. For instance, some barrier systems are designed for large passage ships [20], biogas facilities [47] and operating facilities [51]. In fact, the studies associated with individual barriers continue to increase in some fields, such as road transportation, and the investigation of physical barrier performance is reported by [41,52,53,54]. Therefore, studies on safety barriers can be grouped by two topics: the individual level and the system level, as depicted in Figure 12. Individual-level research is mainly focused on the definition, classification, performance assessment and the principle of individual or technical barriers, while the research on the system level is aimed at issues involved in the barrier system, such as the system design and correlation between different groups of barriers.

### 3.1. Topic 1: Barriers at the Individual Level

#### 3.1.1. Principal Concepts to Describe Safety Barriers

In this section, the basic understanding of safety barriers is addressed, and the contents include but are not limited to, definitions, functions, and classifications. The initiation of safety barriers began in 1973 when Haddon proposed a similar concept for countermeasure strategies against accidents [55]. Later, Reason [14] used the term “*defences*” decomposing into hard and soft defenses, which has an equivalent meaning to barriers. However, Harms-Ringdahl [56] argued that the concept of defense is greater than a barrier; the commonly used hard defenses are regarded as safety barriers that are physical, while soft defenses are beyond the scope of safety barriers, such as regulations, procedures and training. Another important term with a similar meaning to safety barrier refers to the layers of protection analysis (LOPA) proposed by CCPS [57], which stresses the independence between different protection layers [9]. LOPA is widely applied in the oil and gas industry and is presented in both [58] and [59]. In addition, the critical safety element is also functional, similar to safety barriers [60]. According to the definition of safety barriers made by Duijm and Markert [34], safety barriers can be regarded as the aggregation of a series of elements that can be considered safety-critical elements to some extent. Generally, the definition of safety barriers is currently not unanimous and is interpreted in various industries depending on application scenarios. According to the literature available, the various definitions of safety barriers are summarized in Table 3.

Based on the definitions in Table 3, the function of safety barriers can be summarized as “to avoid”, “to prevent”, “to control” and “to mitigate”, which is similar to the discussion in [9,37]. According to ISO 13072 [68], prevention refers to reducing the probability of undesired events, control means limiting the duration of undesired events, and mitigation means lowering the adverse effects of undesired events. To address these functions, different kinds of barriers have been proposed or defined, and the categories of barriers can be determined on the basis of various principles, such as physical or nonphysical [10], functional purpose [40,60], preventive or protective [63], personnel or organizational or technological [64], and static barriers and dynamic barriers [69]. In the present study, according to the classical Bow-tie diagram, all safety barriers are generally grouped into two categories of preventive and protective barriers, which is similar to the definition proposed by Badreddine et al. [63]. The preventive barrier, also known as the proactive barrier, is aimed at preventing the occurrence of incidents or hazardous events or at least reducing the probability of these kinds of events; as a result, the risk propagation can be intercepted. For the other kinds of safety barriers, protective barriers, which are sometimes called reactive barriers or mitigating barriers, are mainly used to alleviate the consequences of incidents or accidents [70]. It is noted that the aforementioned classification of safety barriers is mainly based on one dimension, even though in the classification made by Sklet [9] the barriers related to human/operational are given as double attributions of being passive and active. In a recent study conducted by Sobral and Soares [10], safety barriers were exploratively classified by a classification matrix considering two dimensions, namely, operational types and modes of barriers.

#### 3.1.2. Performance Indicators for Safety Barrier Evaluation

The function of safety barriers is closely related to barrier performance; therefore, the issues associated with the assessment of safety barrier performance are widely considered in the fields of industry and academia. In the implementation of the aforementioned ARAMIS project [37], safety barrier performance is assessed by three criteria, namely, effectiveness, response time and level of confidence. Later, Hollnagel [66] proposed several potential indicators to evaluate safety barrier performance, which may be referred to for specific application scenarios. Janssens et al. [71] developed a decision model by assessing protective safety barrier performance to allocate the barriers correctly against domino effects; meanwhile, the performance of safety barriers to prevent the evolution of domino events was also evaluated by [18]. Later, Landucci et al. [49] proposed a series of key performance indicators to analyze the role of safety barriers in the prevention or mitigation of domino events. To evaluate the performance of safety barriers in the oil and gas industry, Johansen and Rausand [72] proposed several points that can be used to conduct a barrier performance assessment, and the performance of safety barriers designed for gas drilling operations [73], offshore installations in harsh environments [46], the functioning of slug catchers [74] and offshore drilling blowouts [75] are evaluated. Inspired by the work conducted by [37,76], similar indicators to assess the safety barrier performance were designed for Natech scenarios, and the same indicators were also applied by Misuri et al. [21,65] to analyze the performance degradation of safety barriers and the role of safety barriers in mitigating domino scenarios caused by Natech events. The indicators or aspects used to perform an assessment of safety barriers are summarized in Table 4, and their application scenarios are also involved.

According to the existing literature, many methodologies have been proposed to evaluate safety barrier performance by a limited number of parameters or indicators, as shown in Table 4. Some parameters can be observed in different scenarios, such as effectiveness, availability and reliability, while some parameters are applicable for specific applications, such as economic impact and degree of confidence. In the present study, common parameters, including effectiveness, availability and reliability, are discussed in detail.

(1) Effectiveness is widely accepted as a fundamental indicator to assess safety barrier function [37,64,76]. Kang et al. [64] defined effectiveness as an indicator to determine whether a safety barrier prevents accidents, based on which the effectiveness can be assessed by combining professional expert consultation and on-site test data. The study conducted by Kang et al. [64] was mainly aimed at preventive safety barriers. The effectiveness of the protective or reactive safety barrier was defined by Khakzad et al. [78] as an indicator to evaluate the ability to mitigate the damage in the case of a domino event caused by fire. In the case of no consideration for barrier classification, Landucci et al. [18] proposed the hazard intensity reduction factor to quantify the effectiveness of safety barriers, and Misuri et al. [76] defined effectiveness as the possibility that the safety barrier performs well in escalating prevention from a probabilistic-based perspective. In some studies, other terms are used to express a similar meaning of effectiveness; for instance, Hollnagel [66] used efficiency to describe how well the barrier meets the intended purpose, and Shahrokhi and Bernard [79] introduced a function of insufficiency to assess the ability of barriers to impede hazardous events. In practice, the effectiveness of safety barriers is closely related to the duration of the objective barriers being functional after the occurrence of accidents or hazardous events [27], as discussed in the ARAMIS project [37]. In fact, the duration time of safety barriers has been considered while evaluating the effectiveness of the barriers; therefore, in many cases, the duration time or response time is not listed as an independent parameter for safety barrier performance evaluation, such as in [18,49].

The methodologies determining the effectiveness of safety barriers are mainly based on the performance data of the system comprising the objective barriers, and in most cases, operational management, system statement and maintenance are also involved [80]. According to [76], the effectiveness of safety barriers can be determined by
(1)$$\eta_{j,i}=\left\{\begin{array}{l} \eta_{0,i}\;\;\;\;\;\;\;\;\;\;\;\;for\;active\;barrier\\ (1-\phi_{j,i})\eta_{0,i}\;\;\;\;for\;passive\;barrier \end{array}\right.$$
where $\eta_{j,i}$ represents the effectiveness of the ith safety barrier against the ith hazardous elements and $\eta_{0,i}$ denotes the baseline value for the active barrier effectiveness, which is independent of the specific hazardous scenario. In addition, a modification factor named $\phi_{j,i}$ is introduced to characterize the influence of hazardous events on the integrity of the safety barrier with $\phi_{j,i}\in [0,1]$. According to Equation (1), the appearance of hazardous elements has a negligible effect on the effectiveness of active safety barriers, aimed mainly at preventing the occurrence of hazardous events, while passive safety barriers are aimed at mitigating the consequences of hazardous events, and barrier effectiveness decreases linearly with $\phi_{j,i}$. The determination of baseline values for safety barriers presented in Equation (1) can be found in [18,81], who proposed a general procedure for the calculation of $\eta_{0,i}$.

(2) Availability is another widely accepted parameter to describe safety barrier performance, especially for barriers that are active, as discussed by [76]. According to [27], availability can be defined as the capacity of safety barriers to fulfill their anticipated function at a certain time, and the measurement can be made by observing whether the safety barriers have a response when demanded [46,66]; as in the case of the IEC standards [82,83], the average availability refers to the probability of SIS to perform the required SIF within a specific period of time. Sometimes, the availability of safety barriers is expressed by other terms; for instance, in the ARAMIS project, the definition of “level of confidence” is proven to be in line with availability, as previously defined [37].

The availability of safety barriers is greatly affected by the environment, especially in extreme conditions [84], and availability is usually expressed by means of probability. In this case, the probability of failure on demand (*PFD*) is widely utilized to describe the possibility that the system comprised of barriers is unavailable when expressing the safety function is required [76]. The *PFD* value of the safety barrier is largely determined by the architecture of the objective system, and in most cases, the *PFD* can be obtained by standard reliability techniques with sufficient technical data [65], such as fault tree analysis [18], even though in the case of a lack of data, the *PFD* value may also be determined through simplified risk-based methods proposed in IEC 61,511 and 61,508. According to [85] and [86], the PFD values for the safety barriers can be calculated by
(2)$$PFD_{j,i}=1+(\varphi_{j,i}-1)(1-PFD_{0,i})$$
where $\varphi_{j,i}$ is introduced to determine the specific value of $PFD_{j,i}$ on the basis of baseline $PFD_{0,i}$, and $\varphi_{j,i}$ is usually valued by $\varphi_{j,i}\in [0,1]$. According to [76], $\varphi_{j,i}$ is named a performance modification factor that can be obtained by means of professional expert elicitation. In addition, the calculation techniques for the baseline *PFD* value vary depending on the specific scenarios; for instance, in the case of technical data associated with an objective system being available, the baseline *PFD* can be determined by statistical-based approaches [10], and in the case of a lack of available data, expert elicitation may be applied [76], while several traditional probabilistic-based techniques, such as fault tree analysis, can also be utilized to the desired baseline *PFD* [18]. It should be noted that the failure of safety barriers can be caused by various factors. As a result, Sobral and Gudeds Soares [10] argued that the *PFD* of a safety barrier is determined by the sum of the *PFD*s obtained for potential subsystems, such as sensor subsystems, logic subsystems, and actuator subsystems.

Generally, the definition of reliability or robustness is closely related to availability, and in most cases, the availability of safety barriers under fluctuating conditions or assumptions changes when referring to the reliability of objective safety barriers [66,73]. A safety barrier can be regarded as robust or reliable when it is able to withstand extreme or unexpected conditions [61], and the robustness of a safety barrier can be assessed using the variation of availability or effectiveness of the barrier in case the conditions are different [27]. Therefore, in most of the existing literature, the reliability or robustness of safety barriers is frequently considered to be closely related to availability or effectiveness; as a result, they are rarely assessed quantitatively. Coincidently, some concepts proposed to describe or qualify safety barrier performance are also rarely assessed independently, such as the degree of confidence [64], response time [37], resource needs and independence [66]. In many cases, these factors are considered when quantitatively determining the effectiveness or availability of safety barriers.

#### 3.1.3. Modeling Methodologies for Safety Barriers

(1)Modeling safety barrier performance evaluation

According to the existing literature, the studies associated with safety barrier performance assessment at the individual level are mainly concentrated in the field of SIS systems, domino effect events and Natech scenarios. Even though different application scenarios are used, the aspects or indicators used for the assessment are mainly limited to the aforementioned effectiveness and availability. Many methodologies are applied to evaluate the safety barrier performance according to the existing literature, for instance, the well-developed framework of LOPA is aimed at assessing barrier performance by analyzing the independent protection layers [57]; however, the effectiveness of barriers is not considered in the standard LOPA procedure [87]. Another widely applied methodology in the field of safety instrumented systems refers to the safety integrity level (SIL), based on which the international electrotechnical commission (ICE) developed a series of industrial standards, such as IEC 61,508 [85] and IEC 61,511 [88]. Later, on the basis of the integration of principles involved in SIL and IEC 61508, a further comprehensive approach was proposed in the ARSMIS project, which is known as the identification of reference accident scenarios (MIRAS) [37,39].

More recently, the availability-effectiveness methodology was proposed by Landucci et al. [18] for the specific framework of domino effect mitigation, which is largely influenced by the presence and performance of safety barriers [71]. According to the results of the co-authorship analysis implemented in Section 2.2, in the cluster established by Valerio Cozzani and Gabriele Landucci (listed as the most productive scholars, as shown in Figure 3 in Section 2.2), the safety barrier performance is mainly assessed on the basis of two parameters, namely, effectiveness and availability, by the combination of different types of gates, which are presented in Table 5.

In Table 5, PFD represents the value of availability, and the effectiveness of safety barriers is denoted by *η*, while $P_{d}$ refers to the probability of equipment failure. The different types of safety barriers are integrated together by the basic principle of fault tree analysis or event tree analysis, and then the safety level of the system can be quantified. Currently, based on the gates defined in Table 5, events associated with the domino effect are modeled and investigated, such as fire escalation probability assessment for LPG storage [18], fire escalation occurring in an offshore platform [49], domino scenarios in process facilities [89], safety barrier performance for prevention of cascading events in oil and gas offshore installations operations [46], and the performance of safety barriers in the mitigation of domino scenarios caused by Natech events [65].

(2)Modeling safety barrier degradation

Harsh or adverse conditions or events may deteriorate the performance of safety barriers or critical safety elements, regardless of whether they are active or passive [46]; as a result, the protection provided and the possibility of preventing cascading effects may be reduced [21]. Although it is generally recognized that safety barriers with deteriorating performance markedly increase the likelihood of an accident [65], to date, no methodologies have been proposed to accurately quantify the degradation mechanism of safety barriers. However, many existing studies may be referred to for the exploration of these methodologies. The degradation of safety barriers can, to some extent, be indicated by the status of objective barriers, which can be quantified on the basis of technical data obtained by regular inspections and measurements [90,91]; nevertheless, most of the traditional techniques adopted for quantitative risk assessment (QRA) neglect the utilization of new knowledge, information and data (KID), such as traditional Bow Ties [92]. Recently, some methodologies have become available to analyze the impact of specificities, environmental conditions and KID on the health state or performance of safety barriers. For instance, expert elicitation is applied to consider factors not accounted for in the technical database [93], a covariate-based model is proposed to consider the impact of harsh environmental conditions [80]. The newly available KID is utilized to analyze the performance degradation of barriers by using a statistical-based dynamic risk assessment [94], a hidden Markov Gaussian mixture model [95] and a time-dependent reliability analysis [74]. Inspired by the abovementioned studies, more recently, the performance degradation of safety barriers was investigated quantitatively by [21,74] based on the perspectives of multilevel quantification of barriers and multistate Bayesian inference, respectively. The general principle for these two methodologies is compared and illustrated in Figure 13.

Based on the framework illustrated in Figure 13a, this methodology is proposed under the assumption that the performance degradation of safety barriers is mainly caused by the occurrence of hazardous events. The baseline performance of safety barriers is initially assessed by means of the tailored LOPA approach with consideration of factors associated with available technical information, such as maintenance, operational conditions and running data [80], and then, a three-level methodology is proposed on the basis of the uncertainty associated with hazardous events. In the case of low uncertainty of hazardous events, the values for the availability and PFD are regarded as Boolean varieties, namely, they are valued as 0 or 1. A level-one assessment was implemented with increasing uncertainty. A performance modification factor was introduced to modify the values of availability and effectiveness, and this factor can be determined through expert elicitation on the basis of available information on site [76]. If the uncertainty increases further, the L-2 assessment would be applied to identify the modified performance of the safety barrier by means of FTA. Different from the three-level methodology proposed by Misuri et al. [21], Dimaio et al. [74] did not emphasize the calculation of the baseline performance of the safety barrier, as shown in Figure 13b, they paid more attention to the barrier performance variation caused by the varying conditions. The foundation for the study of Dimaio et al. [74] is the assumption that the health state (HS) can be utilized to represent the safety barrier performance in the aspects of judging whether the designed barrier function is fulfilled or not. The barrier HS may be discretized at different levels, such as high, medium, and low. In this way, the safety barrier performance can be regarded as being valued by discretized variables, which are defined between the Boolean variables and continuous variables (the value of performance modification factors) proposed in Misuri et al. [21]. According to [74], the safety barrier HS can be determined on the basis of updated KID in the aspects of key performance indicators (KPI) by quantitative or qualitative approaches. Finally, the safety barriers are mapped into a multistate Bayesian network based on whether the performance variation of the safety barrier can be reflected by means of the variable value of HS.

### 3.2. Topic 2: Barrier Management Level

The design of safety barriers theoretically prevents hazardous events and mitigates the consequences of accidents; however, the roles of various barriers are not correspondingly systematic and stringent in practice. As a result, accidents still occur even though safety barriers exist [96]. An example is the Macondo blowout in 2010, which is attributed to the failure of multiple barriers due to a lack of system barrier management [97], following which Norway issued a guideline on safety barrier management in 2013 [62]. Therefore, in the present study, we will focus on barrier management in the following three aspects.

#### 3.2.1. Basic Principles of Barrier Management Systems

From the engineering perspective, in most cases, the safety of a system or infrastructure is successfully maintained with the comprehensive application of various barriers, as discussed by Kjellen [98]. The design and implementation of safety barriers are considered at the system level, and safety barriers can usually be hardware, software, operational or organizational, which interact with each other [31,73]. As a result, the concept of barrier management systems can be developed. According to the definition by PSA [62], barrier management refers to “coordinated activities to establish and maintain barriers so that they maintain their function at all times”. Later, PSA [99] suggested that the industry should acquire a better understanding of operational, organizational and technical safety barriers and their interactions. Therefore, in the present study, as illustrated in Figure 14, the safety barrier management system is described and reviewed from the following three aspects: barrier element identification, barrier management system and barrier management evaluation.

(1)Barrier elements

The safety barrier system, as illustrated in Figure 14, can usually be broken down into barrier elements that prospectively function to intercept the possibility of risks or threats before and after the identified hazardous events. Conceptually, safety barriers are closely related to layers of defenses, which are widely thought of in terms of the “Swiss Cheese” model [100]. With the initiation of barrier element identification, the potential hazardous events involved in the system need to be identified. For this purpose, typical quantitative risk assessment (QRA) is frequently considered in the development of comprehensive methodologies. The baseline risk assessment tool (BART) is a practical example comprising simplified QRA-related approaches to identify the potential hazardous events arising from the process of oil and gas installations [101]. In addition, vulnerability models designed for infrastructures or equipment are frequently integrated into QRA procedures to determine hazardous events after natural disasters, such as earthquakes [102], floods [103] and lightning strikes [104]. More recently, traditional QRA approaches have been mapped into advanced risk assessment techniques, such as Bayesian networks [105,106], artificial neural networks [107] and directed complex networks [108], to identify and evaluate hazardous events leading to the occurrence of accidents. Based on the identified hazardous events, the barrier elements are mainly determined by means of qualitative approaches, and many graphical techniques are reported to illustrate the identification process of safety barriers, such as event tree analysis [109], safety barrier diagrams [34] and Bow-tie diagrams [77]. These techniques are compared by Sklet [110], who finds that the Bow-tie diagram is the most commonly used graphical technique. In a typical graphical representation of the Bow-tie, the central event is described using several terms, such as top event [111,112], critical event [37], intermediate event [34] and hazardous event [16,75], and in this paper, the term hazardous event is used thereinafter. The left part of hazardous events can be analyzed by fault tree analysis (FTA), while the right part can be coped with event tree analysis (ETA) [113], by which barrier elements are identified for different application scenarios [37,63,74,75].

To date, the definition, function and classification of barriers or barrier elements have been studied extensively, as discussed in Section 3.2.1; however, the criterion to be a barrier element and the performance requirement for a standard barrier need to be investigated and discussed further. Currently, there is no clear distinction between safety barriers and other terms associated with safety, such as safeguards, safety measures and countermeasures, especially for human-related barriers. As a result, the Center for Chemical Process Safety [114] and the Chartered Institute of Ergonomics and Human Factors [115] argued that most human-related measures should be treated as safeguards rather than barrier elements. This may be because most of the barrier elements are determined based on barrier function while ignoring the working principle of barrier elements. The criteria and performance requirements of physical or technical barrier elements are easily obtained by specific scenarios, the experience of professional experts and the available technical data, such as the barrier elements involved in barrier-based models for drilling blowouts [60], barrier systems designed for leakage in oil and gas production [116] and safety barrier systems for hydrogen refueling stations [117]. However, the human-related barrier elements in the organizational and operational aspects are more complex and difficult to describe. Many scholars try to determine and develop human-related barrier elements from the perspective of safety management. Most of the studies are implemented by qualitative approaches. For instance, King et al. [20] designed barrier systems involving organizational and operational barrier elements to maintain the stability of large passenger ships. Bucelli et al. [25] described a barrier system associated with human-related barrier elements for safer operation in the oil and gas industry. Nevertheless, the CIEHF [115] proposed a general performance standard for human-related barrier elements with coverage of at least seven aspects, and the performance criteria for these barrier elements were also involved.

(2)Barrier management system

According to PSA, barrier management has been regarded as the main priority because accident investigation clearly indicates that the failure and weakening of barrier elements are the principal contributing factors to accidents [96]. There is no doubt that various safety barriers should be systematically implemented in a consistent manner to minimize risks. Although Harms-Ringdahl [56] argued that safety barriers should be limited to technical or physical barriers based on the perspective of the layer of defense, it is widely accepted that software, especially some human-related safeguards, should be involved in barrier systems. As Øie et al. [16] and Lauridsen et al. [96] discussed, the integrative safety barrier system should be comprised of at least three different kinds of barriers, namely, technical, operational and organizational barriers, as shown in Figure 11. Practically, the subsystem comprised of technical or physical barriers is frequently studied in various scenarios, and the interactions between individual barriers are also presented by means of probabilistic-based techniques or fuzzy-based approaches. Based on the classification of the application scenarios of the safety barriers by different industries, it is interesting to find that the barrier management mode is nearly unique for a certain industry. For instance, in the chemical industry, barrier management emphasizes the integrity of different technical or physical barriers, although in many cases, these barriers are presented as subsystems, such as in [9,45,71,117,118]. A similar phenomenon can also be observed in the field of natech scenarios [21,76]. However, in the field of offshore oil and gas, some of the studies are similar to the work conducted in the chemical industry, e.g., barrier management is focused on the combination of various technical or physical barriers. Most of the remaining studies pay more attention to the role of human-related barriers, mainly referring to operational barriers and organizational barriers. Especially for projects implemented by PSA [62] and DNV GL [16], operational and organizational barriers are given equal consideration as technical barriers. In fact, as early as 2006, the concept of a barrier integrated set (BIS) was proposed by Miura et al. [119] to comprehensively consider the role and interaction of various barriers. Later, Pitbaldo and Nelson [120] included human and organizational aspects in barrier management, and Lauridsen et al. [96] tried to further investigate the interaction between technical, operational and organizational barrier elements. In addition, the failure of human-related barriers can be evaluated quantitatively by human reliability assessment (HRA) with reference to [121]. In the maritime shipping industry, King et al. [20] designed stability barrier management for large passenger ships based on the studies implemented by [16].

The practical activities associated with safety management have proven that all the barrier elements are related to human factors [122], and in a typical safety management system, the factors stemming from social and technical fields are influenced by each other; therefore, it is necessary to study the barrier management system from the perspective of complex socio-technical systems. In a typical complex socio-technical system, humans are widely accepted as the most positive element and assuring the reliability of human-related barrier elements is critical for the performance and function of the designed barrier management system [50]. Achieving the true independence of technical barriers in terms of their reliance on organizational or operational barriers may be challenging. Unfortunately, although the importance of the intersection and interaction between technical barriers and human-related barriers has been recognized by some scholars, such as [10,34,122]; few studies associated with interaction issues have been reported according to the existing literature. However, some explorative studies may be helpful for the investigation of these issues. Some influencing factors for safety barrier performance identified by Prashanth et al. [73] can be classified as human-related barriers by the identification principle proposed by CIEHF [115]. In many cases, the influencing factors are also known as risk influence factors (RIFs), whose relationship with barriers may be analyzed by a barrier model and operational risk analysis (BORA) proposed by Aven et al. [38]. Later, an extension of the BORA model named risk-OMT was proposed by Vinnem et al. [123] to further identify the RIFs considering the decomposed operational barrier functions.

(3)Barrier management evaluation

According to the perspective of safety management, the design and implementation of safety barrier management systems should be an integrated part of safety management [72]. As illustrated in Figure 11, the barrier management evaluation is played as feedback for the improvement of barrier element identification and safety barrier system design. Generally, in this paper, the issues of barrier management evaluation are reviewed in the following two aspects, namely, dynamic barrier management and the contribution of barrier management to safety management.

After the establishment of a barrier system, dynamic barrier management should be developed and implemented because the performance and function of barrier elements involved in this system may be degraded or influenced by external environmental conditions and the internal factors within the barrier elements themselves. Essentially, dynamic barrier management is aimed at preventing the degradation of barrier elements and repairing degraded barriers [69]. For this purpose, in the offshore oil and gas industry, some companies develop and implement their own safety management programs that function similarly to the aforementioned dynamic barrier management, such as the manual of permitted operations (MOPO) and the tripod investigation approach pioneered by Shell [69,124], the performance monitoring approach adopted by BG [125], and the technical integrity management program (TIMP) initiated by Statoil [126]. More recently, DNV GL proposed a dynamic barrier management program with the objective of blowout prevention [127]. Perrin et al. [128] proposed a methodology named Method Organized and Systemic Analysis of Risk (MOSAR) or Analysis of Dysfunctions of the Systems (MADS) to improve the performance of normative barriers. According to the study conducted by Pitblado et al. [69], dynamic barrier management comprises the following stages: data collection from multiple sources, prediction of barrier status, impact evaluation of barrier status onto risk and finally, decision support analysis. Therefore, the key to dynamic barrier management is the performance monitoring and prediction of the barrier elements involved in the barrier systems. The issues of barrier degradation have been discussed in Section 3.1.3, in which the roles of humans and organizations are given less attention. Pitblado and Nelson [120] proposed a comprehensive methodology that integrates barrier-based risk assessment and “success pathways” with full consideration of the positive roles of humans and organizations.

The popularity and acceptance of safety barriers and barrier management in both industry and academia are mainly due to their applicability in risk reduction and accident prevention. Barrier management can certainly be regarded as one of the advances in the field of safety management, and barrier-based diagrams have proven to be a useful tool in documenting safety measures adapted to prevent accidents [34]. As the critical component in safety management systems, barrier management functions to control risks and acts as the input of the system [129]. Therefore, traditional safety management audit assessment approaches can be used to maintain the reliability of safety barriers [130], such as the I-risk management audit technique [131] and ARAMIS audit methodology [132]. Duijm et al. [133] complemented barrier-oriented audit protocols with the implementation of safety culture questionnaires. According to [32], accidents that occurred in the Netherlands ranging from 1998–2004 were constructed by a software tool, storybuilder, developed within the framework of Bow-ties, based on which the success and failure modes of safety barriers were identified and analyzed to optimize the control of occupation risks. Later, Bellamy et al. [134] found that the failure of safety management is mainly due to a poor understanding of the motivation and awareness of safety barriers. In France, the National Institute of Industrial Environment and Risks (INERIS) regards safety barriers as an important tool to implement risk control [33]. Chen et al. [45] integrated security measures, safety barriers and emergency responses into a comprehensive model named the dynamic vulnerability assessment graph to manage the human-related domino effects in chemical industrial parks.

#### 3.2.2. Typical Application of Barrier Management Projects in Practice

The potential of safety barriers to manage risk allocation before and after accidents is developed and put into industrial practice in the form of safety-oriented projects. In this paper, these projects are reviewed and analyzed hereinafter.

(1)Accidental risk assessment methodology for industry projects.

The ARAMIS project was co-funded by the European Commission with the objective of satisfying the requirement of the SEVESO II directive. This three-year project was launched in January 2001 and ended in 2004. One year later, the methodology proposed in the project was applied in the industry. Within the ARAMIS project, there are mainly six steps involved in implementing the risk assessment in the decision-making process [39].

The first step is to identify all the major hazardous events involved in the process industry, during which the Bow-tie diagram is developed with the integration of fault tree analysis and event tree analysis. In most cases, the identification of critical events for specific scenarios is emphasized in this step, and many probabilistic-based methodologies can be utilized here.

The second step focuses on the identification of safety barriers. In this stage, the safety barriers are defined by their function, performance, classification, and level of confidence. Notably, the performance monitoring and assessment of safety barriers are considered important and need to be studied.

The third step is to evaluate the safety management efficiency to barrier reliability. Within the ARAMIS project, the existing safety management system and safety culture are assumed to influence the reliability of safety barriers; therefore, a process-oriented audit protocol is embedded in the ARAMIS procedures to review the activities relating to safety barriers.

In the fourth step, the reference accident scenario (RAS) is defined and identified. Usually, the RAS refers to the initiating events that cause critical events; in some cases, the terms trigger events are also used to describe the RAS. The specific severity index for RAS can be quantified with reference to [135].

The fifth step is to map the risk severity of reference scenarios based on the results of risk severity assessment. Risk severity is represented geographically by a combination of the frequency level and intensity effects. Finally, risk severity can be mathematically calculated by multiplying the frequency and severity index obtained in the fourth step.

The last step in ARAMIS is to evaluate and map the vulnerability of the environment independently of hazardous events, which is beneficial for local authorities to take measures to reduce the global risk level, perhaps neglected by the operator on site. Global vulnerability is actually a linear combination of each target vulnerability, which is determined by the concerns of all stakeholders.

(2)The barrier management project launched by DNV GL.

Almost at the same time as initiating the ARAMIS in 2001, the DNV GL collaborated closely with Statoil to implement a program named the technical condition of safety barriers (TTS), which is mainly aimed at monitoring the identified key safety barriers [69]. In the TTS program, all the critical safety barriers are evaluated in terms of their original design, conditions and operation, which are scored “A–F”. In 2010, another program named the technical integrity management program (TIMP) was launched by DNV GL to implement risk control in conjunction with TTS [126]. Recently, DNV GL published the guideline “*barrier management in operation for the rig industry—good practices*”, which aims to increase the understanding of barrier management at the management level and operational phase for both onshore and offshore [16]. In this guideline, the establishment and implementation of barrier management are described in detail. Later, DNV GL released a QHSE software solution for barrier management named Synergi Life on the basis of a Bow-tie model. With the application of Synergi Life, missing and degraded barriers can be effectively identified and monitored, and other operational elements can also be embedded conveniently depending on the requirements. In addition, the barrier management project of DNV GL also covers the fish farming industry, which supports the sustainable development of fish farming by improving the operational risk management level. Another barrier management program proposed by DNV GL refers to MyBarrier with the objective of applying it in offshore oil and gas industries. MyBarrier is able to quantitatively assess the impacts of component failures on the risk of losing a barrier element by using real-time data and information.

(3)Standards (generic and industry) and guidelines associated with barrier management.

The development and application of safety barriers are taking place continuously at a rapid pace, and the objective of barrier management is to harmonize the various safety barriers in an orderly way; as a result, the anticipating functions of barrier systems can be successfully maintained. For this purpose, many nonprofit organizations and authorities, including but not limited to standardization organizations, industrial associations, industrial committees and industrial authorities have offered standards and guidelines, both generic and industrial, in recent decades. These standards and guidelines are beneficial for practitioners in terms of barrier application and management. In this paper, some of the standards and guidelines associated with barrier management are listed in Table 6.

It can be seen from Table 6 that the standards and guidelines associated with barrier management are mainly concentrated in the oil and gas and process industries. In the oil and gas industry, a great contribution is made by Norsk Sokkels Konkuranseposisjon (NORSOK) and PSA from Norway. In the publication of NORSOK [136], a total of 20 barrier systems are listed that can be decomposed further into various barrier elements. Later, PSA issued a guideline on the management of barriers [62] based on the basic principle of the Bow-tie diagram. In 2016, a guideline titled “*Guidance for barrier management in the petroleum industry*” was published by Hauge and Øien [137] from SINTEF. At almost the same time, the Center for Chemical Process Safety in the U.S. also offered guidance for the management of barrier elements from the perspective of Bow-tie diagrams [114]. In the process industry, most of the contributions associated with barrier management are attributed to the international standard organization (ISO) and the international electrotechnical commission (IEC), both of which issued a series of barrier management standards, including generic and industrial standards. For instance, the IEC issued a series of standards [82,83,138] to guide the management of safety-instrumented systems that essentially correspond to barrier systems. In addition, the role of human-related barriers, including operational barriers and organizational barriers, is a concern of the Chartered Institute of Ergonomics and Human Factors (CIEHF), which published the “*Human Factors in Barrier Management*” guideline. The highlights in this publication are represented by (1) the proposed principle to determine whether a safeguard is a safety barrier element, (2) the performance standards for human-related barriers, and (3) the management procedures designed for human-related barriers.

#### 3.2.3. Issues Discussed for Barrier Management

Although there are many documents associated with safety barriers and barrier management, such as standards, guidelines, reports, and research papers, many challenges still exist during the implementation of safety barrier frameworks in practice. Across various scenarios, different barrier-related aspects can hardly maintain consistency; for instance, barrier strategies, performance requirements, and operational procedures, in many cases, cannot harmonize comprehensive integration according to the audit results reported by PSA [144]. Therefore, it is necessary to discuss the issues that need to be studied for the implementation and improvement of barrier frameworks.

(1)Issue 1: Lack of clear clarification for the boundary of safety barriers

According to the existing literature or technical reports, nearly thousands of safety barriers or barrier elements have been proposed and defined [69]. However, where is the boundary of safety barriers? Few studies have focused on this issue, and in most cases, the boundary between safety barriers and safeguards or safety measures is not clear, especially for human-related barriers, such as the operational barriers and organizational barriers defined by DNV GL [16]. Furthermore, in some studies, human-related or organizational factors are identified as performance influence factors (PIFs) for technical or hardware barriers [73,96,137,145] instead of barrier elements, and studies on these factors are aimed at providing flexible conditions under which technical barriers are able to function as expected. Another important area where there is a lack of clarity is in the terminological inconsistencies between regulations and standards, which makes it confusing during the implementation of barriers [146]; for instance, the terms barriers/barrier elements and barrier performance/status are frequently used interchangeably. Another source of confusion lies in the fact that different guidelines or standards are issued by authorities from different countries; for example, the guidelines recommended by PSA in Norway and CCPS in the U.S. may differ in terms of terminology and principles.

(2)Issue 2: Role of human or organizational barrier elements in barrier management

This issue is discussed under the assumption that human-related barriers are important components in the barrier management system, which is widely accepted at present. First, the distinction between human-related barriers and the PIFs for barriers needs to be clarified further. For instance, safety culture is considered a kind of barrier in some studies; however, is there any interaction between safety culture and other technical barriers? In addition, practitioners on site are frequently confused by the relationship between human-related or organizational safety barriers and standard operating procedures (SOPs). More importantly, in the case of the introduction of human-related barriers into safety management, an important issue that emerges is “how to cope with the relationship between barrier management and human reliability analysis (HRA)?” In this case, some concepts have to be recognized to obtain a better understanding of human-related barriers. For example, human errors or human-related mistakes in most cases are regarded as the causes or trigger events leading to incidents or accidents, as described in many accident models under the HRA framework; however, from the barrier management perspective, human errors are considered the consequence of human-related barrier failure, that is, human errors are results rather than causes. Unfortunately, to date, few studies have investigated the functioning of human-related barriers with reference to the HRA framework.

The ambiguous understanding of human-related barriers is essentially determined by the difficulties of describing these barriers in the aspects of function, performance and the monitoring approach applicable for specific scenarios, especially in the case of the unavailability of required reliable data. Although CIEHF [115] tentatively proposed a framework to describe human-related barriers, it is still difficult to substantially influence various operational industries. In addition, in the guidelines reported by CIEHF [115], the interaction between human-related barriers and technical barriers is not given much attention, which may be an important issue to solve in the near future. Another particular challenge for the industrial application of human-related barriers lies in the fact that there is a lack of guidance for establishing performance requirements and monitoring procedures, as well as the assessment framework.

(3)Issue 3: Integrating various safety barriers into existing safety management

In the early application of barrier management, barrier elements were generally identified as technical barriers that could be described and evaluated quantitatively. However, the occurrence of accidents is a reminder that accident prevention measures should be comprehensive, especially after the Deepwater Horizon accidents. The Chemical Safety Board argued that with the necessary actions taken, serious consequences may be avoided [147], which explains the importance of barriers associated with humans or organizations. Unfortunately, only a few technical barriers are able to function within limited industrial scenarios, let alone human-related or organizational barriers. The challenges are mainly attributed to the gap between barrier management and safety management in use, even though both are aimed at controlling various risks, and the implementation process is different to a large extent.

At present, the safety management system is running well in most industrial companies, and the safety audit approaches are also standardized. Therefore, a common concern among industrial practitioners is how to map barrier management into the existing safety management system. In practice, it can be reasonably predicted that the introduction of barrier management would increase the complexity of the safety management system. Furthermore, most of the quantitative analysis approaches (QRAs) frequently used in traditional safety management are not applicable for barrier management because most of the nonphysical barrier elements cannot be described quantitatively. Another important concern about this issue lies in the compatibility of barrier management and planned maintenance [146]. As mentioned in Section 3.1.1, the performance of barrier elements deteriorates with time, similar to ordinary mechanical equipment. The latter is practically maintained by establishing the typical planned maintenance system; however, the maintenance of the former is not solved up to date; for instance, how can the test intervals or maintenance period of the barrier elements be determined? In addition, the function of most barrier elements cannot be tested in the simulation circumstance; if the function verification is implemented in the real scenario, the verification activity may induce unexpected failures or accidents. The uncertainty or vagueness of barrier element maintenance would also confuse practitioners with distinctions between system failure and barrier criticality. According to audits reported by PSA, many oil and gas companies are not able to exactly classify failure and barrier element criticality [137].

(4)Issue 4: Dynamic assessment of barrier elements based on a system perspective

According to the definition proposed by PSA [62], the purpose of barrier management is to maintain the function of barriers, which is generally implemented by dynamic assessment of objective barrier elements. The methodologies employed in the existing literature associated with QRA can be consulted to conduct dynamic barrier assessment, such as dynamic Bayesian network and failure tree analysis. However, there are still three important issues that need to be studied further: (1) what is the benchmark for the assessment? (2) How can real-time data and information be obtained and integrated? (3) How can the interaction among the various barrier elements involved in the barrier system be coped with?

It is essential to set a reasonable benchmark for dynamic barrier assessment; however, the determination of the benchmark for various barrier elements is not easy, especially for nonphysical barrier elements. In the guidance provided by PSA [62] and DNV GL [16], the benchmark for dynamic barrier assessment is not explained in detail, which makes it difficult to distinguish the barriers that function or are impaired. Meanwhile, the benchmarks designed for a single barrier element and the barrier system may be different; therefore, it is necessary to understand the relationships among different benchmarks. For instance, before initiating a specific assessment activity, all the performance requirements of barriers should be reviewed and analyzed. The second issue is focused on the technical data and information, current technical developments are trying to make it possible to obtain real-time data [120], and advanced tools are developed and tentatively applied on advanced oil and gas platforms [148]. However, companies will not implement risk control activities at any price; for example, a common complaint about the barrier management requirement proposed by PSA is the significant costs related to functional testing and the establishment of an indicator system [72]. Therefore, how to balance the cost and gains is a challenge for the application of industrial barrier management that cannot be ignored. The last issue involved in the dynamic management of barrier systems emerged from a consideration of the interaction between barrier elements. Generally, the objective of barrier management is supported by integrating multiple barriers; as a result, the interaction between technical, operational and organizational barrier elements should be well understood [96]. For instance, traditional technical barriers frequently act as active barriers or preventive barriers on the left side of the Bow-tie diagram, while operational and organizational barriers are usually applied as reactive barriers or protective barriers on the right side of the Bow-tie diagram. In the state of “work as image”, the positive interaction of both sides of the Bow-tie diagram should be observed in an effective safety management system. Therefore, the interaction between different kinds of barrier elements should be fully considered in dynamic assessment activities for barriers, whether at the single or system level.

## 4. Future Research on Safety Barriers

### 4.1. Correlation and Synergism of Different Safety Barriers within a Complex Socio-Technical System

According to the review of the collected papers, the studies associated with hardware barriers or technical barriers have far exceeded those associated with operational and organizational barriers. There is no denying that technical barriers alone are not able to respond to the expected functions of barrier management by industrial companies. Therefore, the development and improvement of barrier management must be implemented within the framework of a complex socio-technical system in which the role of human-related barriers, such as operational barriers and organizational barriers are given much attention to integrating other kinds of barrier elements. In such a complex system, the human-related components should be reviewed and analyzed thoroughly to determine which act as the safeguards and which can be attributed to barrier elements, based on which the definitions, functions, expectations, performance requirements and assessment approaches of human-related barrier elements need to be described. In addition, at the system level, traditional QRA methodologies may be inapplicable for complex socio-technical systems, and advanced technologies should be considered for barrier management system assessments, such as artificial neural networks, complex networks and data mining techniques. In the process of modeling barrier system assessment, the various barrier elements, including technical, operational or organizational elements, can be regarded as “agents” that facilitate the quantitative or logical description of potential interactions by means of developing interactive rules and algorithms. Meanwhile, the paravirtualization of barrier elements as “agents” is also able to trigger the initiation of positive human-related barrier elements by adjusting the barrier elements themselves to adapt to new or hazardous situations; this may be a potential research perspective of great importance in the near future.

### 4.2. Allocation of Safety Barriers within the Framework of “Risk Capacity”

According to the discussion presented in Section 3.2.3, the integration of barrier management into daily safety management activities is regarded as a salient challenge for the industrial application of barrier management. If barrier management is independent of a company’s safety management, then the outcomes of the program associated with barriers would just collect dust on the shelf. Traditional safety management is generally divided into two parts: the front end and the after end. The front end focuses on the identification and assessment of risks for specific scenarios, while in the back end, the capacity of controlling the identified risks is developed, the whole process of which can be summarized as a “risk-capacity” framework. Then, barrier management can be mapped into this “risk-capacity” framework by bridging the abovementioned front end and after end; as a result, the framework of traditional safety management becomes “risk-barrier capacity”. The salient advantage of “risk-barrier capacity” is that barrier management can be integrated into the existing safety management system with minimum disturbance for daily activities. In addition, awareness regarding barriers can also be understood and accepted well by operators or practitioners, which has a positive influence on critical barrier functions [149]. Within the framework of “risk-barrier capacity”, at the front end of risk assessment, it is necessary to note that human errors may not be identified as risks with the introduction of operational and organizational barriers. In contrast, human errors should be treated as the failure of human-related barriers, which is equivalent to the failure of technical barriers or mechanical equipment. Then, the results of risk assessments act as inputs to barrier management to design barrier systems. At the end of capacity development, the designed barrier system acts as the input to be embedded into the safety management system of companies in which safety culture, operational procedures, etc., may be regarded as operational or organizational barriers.

### 4.3. Response to Challenges Stemming from Industry 4.0 and Intelligence

Industry 4.0, also known as the fourth industrial revolution, mainly refers to the convergence of manufacturing with the digital revolution, artificial intelligence, the internet of things and smart devices [150]. In this era, intelligence plays an increasingly significant role in safety management, which may lead to the concept of safety intelligence [151]. The potential research perspectives of barrier management in the era of Industry 4.0 would be represented by three aspects. 

(1) The first aspect lies in the application of intelligent techniques to barrier management. The most feasible application currently may be the status monitoring of physical barriers by intelligent sensors, which is still in development [152]. Intelligence is also characterized by inimitable advantages in the field of decision-making, which is the core of barrier management. For instance, on the basis of data or information associated with barrier element performance, some intelligent algorithms may be utilized to assess the performance of barrier systems, and then, recommendations for barrier system optimization can be proposed. 

(2) The second aspect emphasizes the application of big data to characterize the barriers. The acquisition of real-time data associated with barriers is critical for barrier management [69]; however, in many cases, the available data for highly reliable systems, including safety barrier systems, are insufficient [153]. Therefore, there will be at least two research perspectives proposed in this paper, namely, data acquisition and data processing technologies. In terms of data acquisition, the technical data associated with physical barrier elements can be obtained by means of advanced technologies, such as the intelligent technology discussed in Section 4.3, while the acquisition of data involved in the human-related barrier elements may not be easy work. For instance, the required behavior records of operators are usually difficult to obtain, for reasons related to safety culture and safety awareness in the company. Another important aspect is focused on the interpretation of the obtained data, which is mainly implemented by the application of big data analytics and the development of data-driven approaches.

(3) The last aspect of the research perspective would focus on the application of barrier management defending the risks presented in the era of Industry 4.0. The risks within Industry 4.0 are characterized by complexity and uncertainty, which requires the comprehensive integration of various barrier elements, and the interaction between different barrier elements needs to be emphasized, as discussed in Section 3.2.3. In addition, in the era of Industry 4.0 characterized by high atomization, the reliability of barrier elements would be considered first, as discussed by Agrawal et al. [154], for the challenges involved in safety-critical intelligent systems. Overall, in the future, the concept of intelligent barriers will be developed; however, there is still a long way to go before the industrial application of intelligent barriers takes place.

### 4.4. Resilience Theory to Enhance Barrier Management

In a broader sense, the failure or performance degradation of components in the complex socio-technical system can be essentially regarded as a kind of abnormal status of the system, which is similar to the viewpoint of safety II [155,156]. The capacity of the system to recover from abnormal status to normal status is usually measured by resilience. Even though there is no universal consensus on the definition of resilience, it is widely accepted that the resilience of a system should be at least represented by three aspects, namely, absorptive capacity, adaptive capacity and restorative capacity [157]. According to the perspective of the Bow-tie diagram, both the absorptive and adaptive capacity take effect on the left side to absorb or adapt the identified risks, while the restorative capacity mainly functions on the right side to facilitate the system recovery from failure status to normal status. Therefore, it is easily observed that the absorptive and adaptive resilient capacities can be developed by preventive safety barriers, and the restorative capacity can be developed by means of protective barriers. Overall, the resilience capacity of a system can be developed by means of barrier management, which provides a potential practical strategy to promote industrial application based on resilience theory. In addition, methodologies for system resilience assessment can also be consulted to assess barrier systems, such as the resilience assessment grid (RAG) [158] and functional resonance analysis method (FRAM) [159], both of which are applicable for complex socio-technical systems. Currently, the development of system resilience is mainly contributed by operational and organizational barriers, and in most cases, the performance degradation of barrier systems is caused by external disruption; therefore, it can be reasonably inferred that the improvement of absorptive and adaptive capacity by allocating preventive barriers may be useful to slow the performance degradation process.

Another important aspect associated with barrier management is the health status monitoring of barriers. Then, early warnings for performance-degraded barrier elements could be given. As an important proposal, data processing algorithms should be embedded in the designed barrier management system. The data processing methodology for barrier element health status monitoring or assessment is expected to be developed based on advanced technologies, such as machine learning, artificial neural networks and data-driven Bayesian networks.

## 5. Conclusions

Safety barrier management is one of few safety management frameworks with an industrial application, including the development of industrial guidelines, standards, and application software. In this article, a review of barriers and barrier management from the perspective of accident prevention was implemented by two different methods, namely, bibliometrics and a systematic literature review that were integrated to investigate the basic principles, advances and research perspectives in the fields of barriers and barrier management.

The main body of this study is represented by three modules. First, the maps associated with barriers and barrier management are illustrated to analyze the various scientific networks, obtaining insights for networking and collaborations for the study of barriers by means of bibliometrics. Then, the advances in safety barriers are discussed on the basis of barrier research topics at the individual level and barrier management level. In this section, six aspects of safety barriers are reviewed in detail. Finally, five research perspectives for safety barriers are proposed. The general idea emphasizes the importance of nonphysical barrier elements, such as operational and organizational barrier elements, which should be studied from the perspective of complex socio-technical systems. As a result, the proposed “risk-barrier capacity” framework in this paper can be developed. In addition, the authors argue that the development of barrier management will benefit from the application of intelligent techniques and the framework of system resilience.

## Figures and Tables

**Figure 1 ijerph-19-09512-f001:**
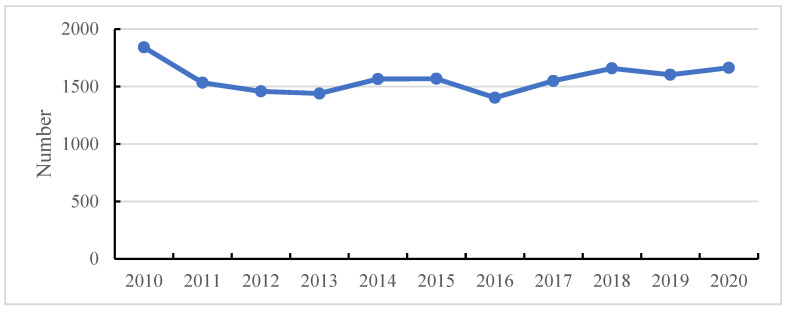
Maritime shipping accidents that occurred in the 2010–2020 period.

**Figure 2 ijerph-19-09512-f002:**
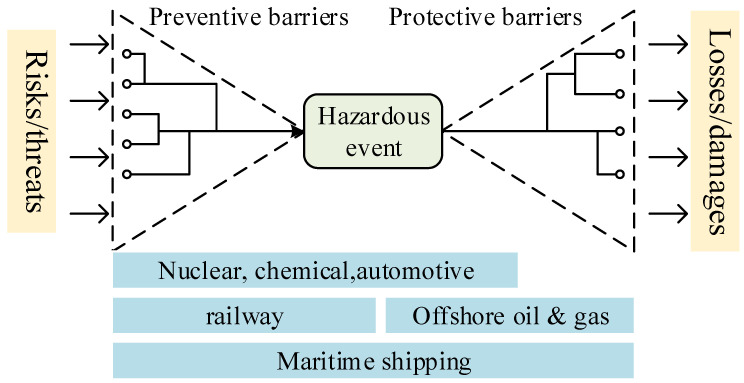
Emphasis of risk control for different industries [8].

**Figure 3 ijerph-19-09512-f003:**
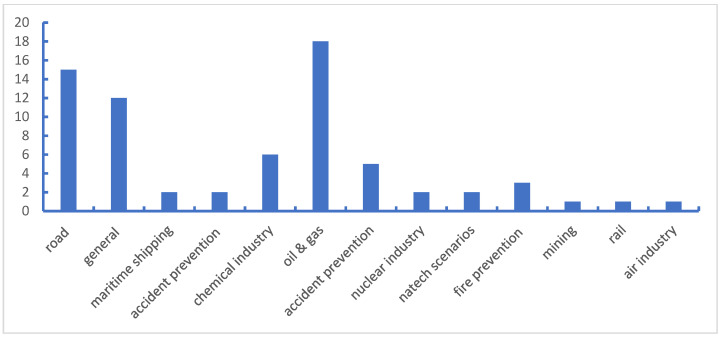
Distribution of the articles involved in the database.

**Figure 4 ijerph-19-09512-f004:**
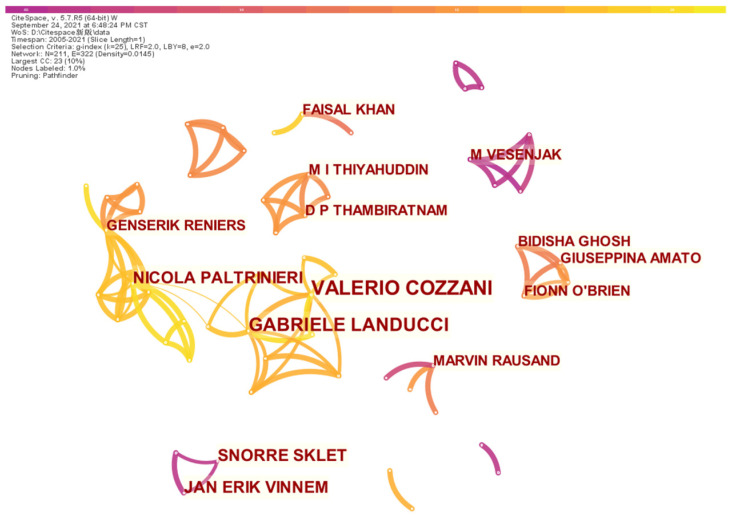
Illustration of scholar collaboration by co-authorship analysis.

**Figure 5 ijerph-19-09512-f005:**
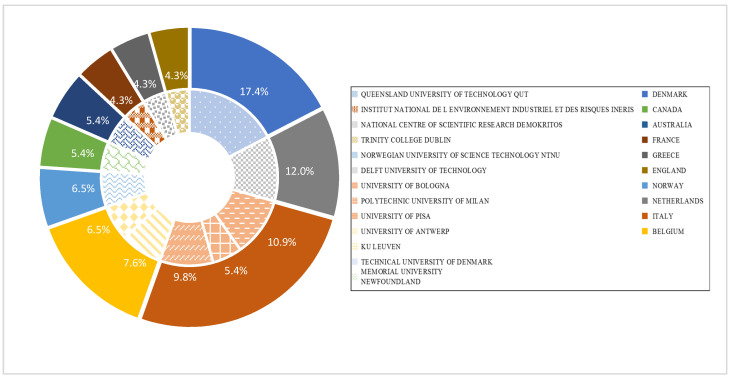
Distribution of authors’ affiliations.

**Figure 6 ijerph-19-09512-f006:**
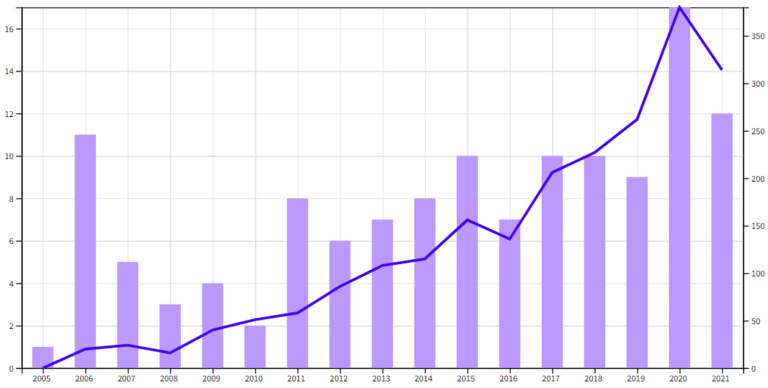
Papers published by year of publication according to the developed database. 
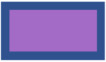
 Publication, 

 citation.

**Figure 7 ijerph-19-09512-f007:**
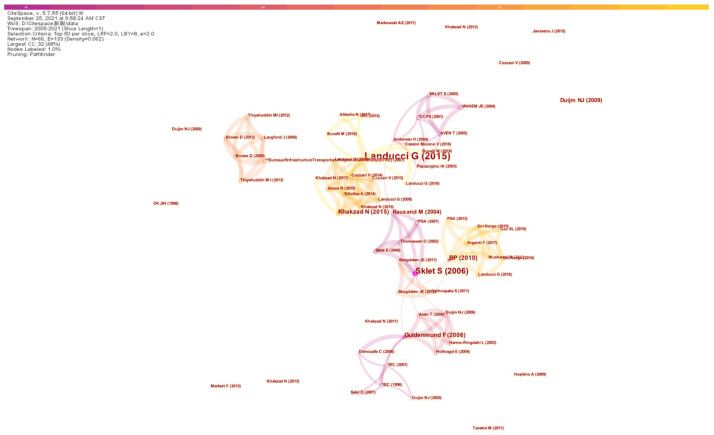
Network of papers-relatedness based on citations.

**Figure 8 ijerph-19-09512-f008:**
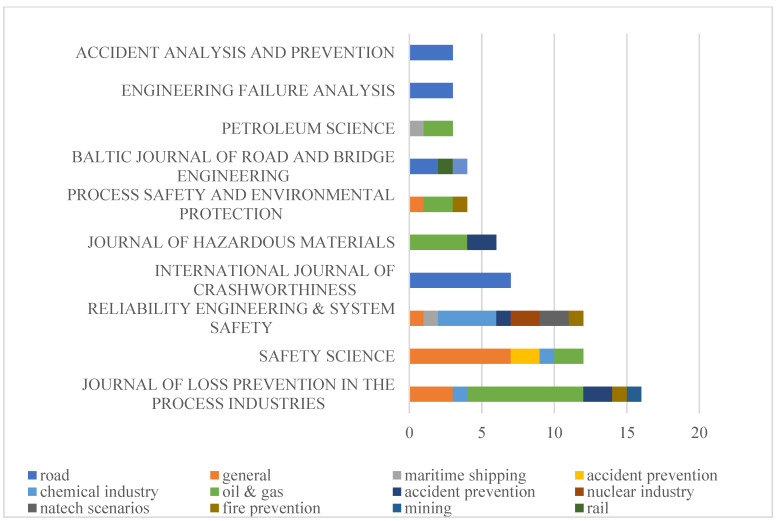
Number of publications by journal.

**Figure 9 ijerph-19-09512-f009:**
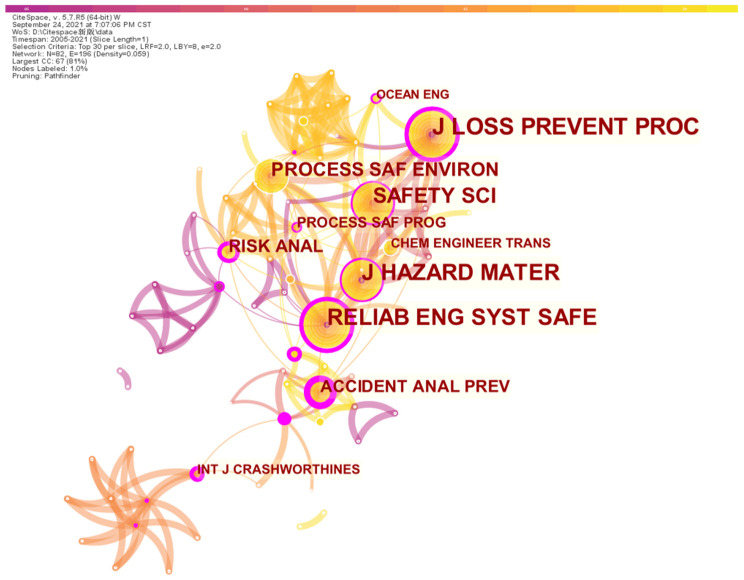
Network of paper sources with regard to co-citations.

**Figure 10 ijerph-19-09512-f010:**
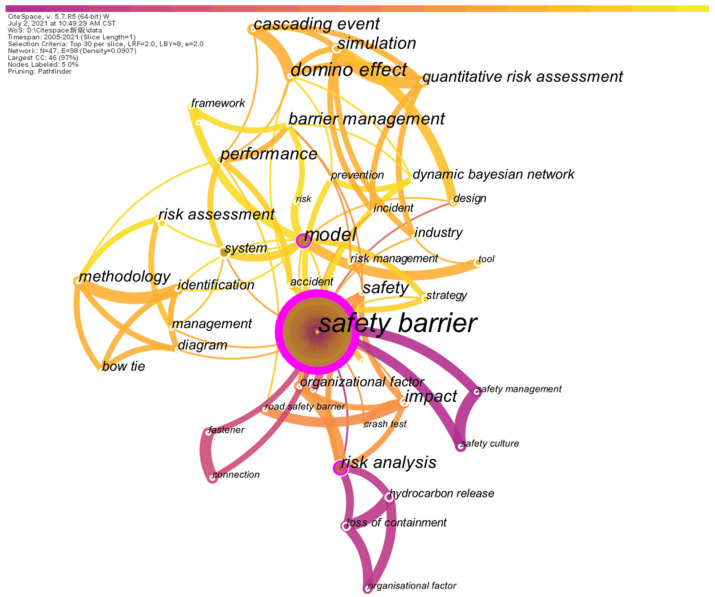
Mapping of keyword co-occurrences.

**Figure 11 ijerph-19-09512-f011:**
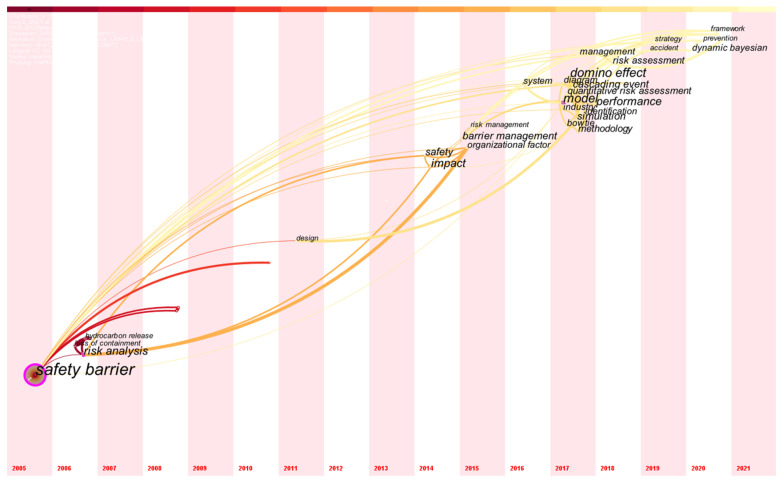
Evolution diagram based on keyword cooccurrences.

**Figure 12 ijerph-19-09512-f012:**
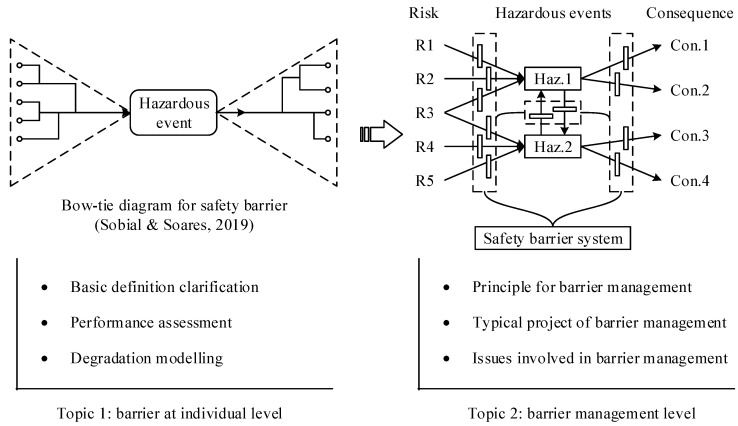
Studies overview of safety barrier issues.

**Figure 13 ijerph-19-09512-f013:**
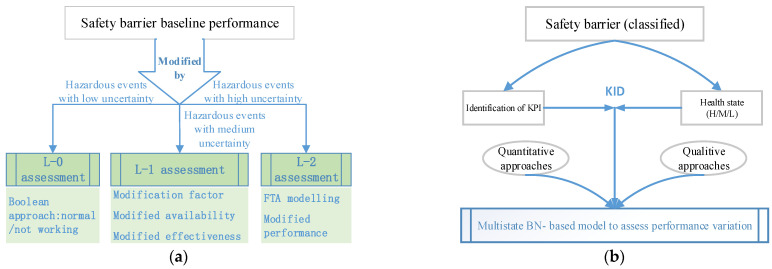
Methodologies to assess the performance degradation of safety barriers. (**a**) With reference to [21]. (**b**) With reference to [74].

**Figure 14 ijerph-19-09512-f014:**
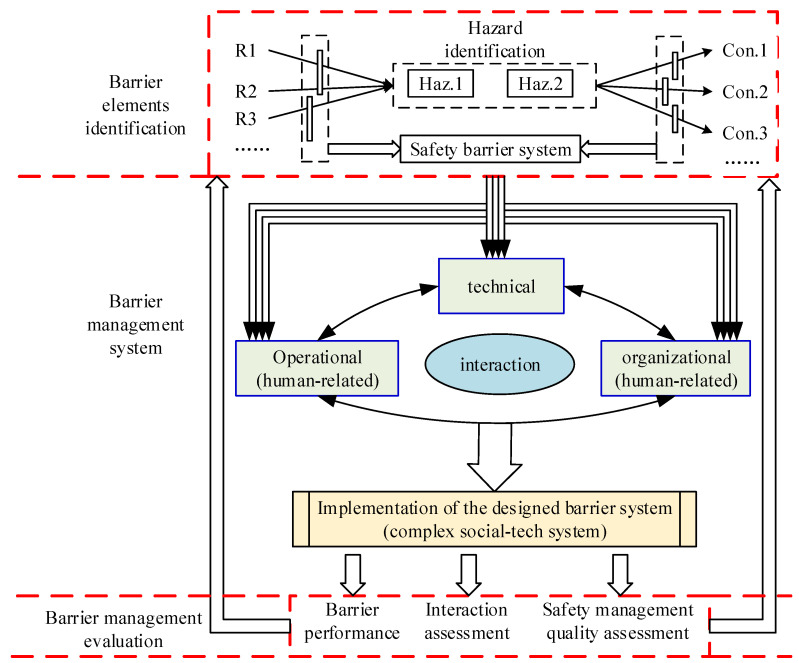
Basic principle of the safety barrier management system.

**Table 1 ijerph-19-09512-t001:** Article collection protocol.

Subject	Description
Databases	Core collection in *Web of Science*
Title keywords	Safety barrier, barrier management
Search field	Title, keywords
Boolean operation	OR
Document type	Research articles
Language	English
Time interval	1978–2021.07

**Table 2 ijerph-19-09512-t002:** Top 10 papers by number of global citations.

Author	Journal	Citations	Application
[9]	JOURNAL OF LOSS PREVENTION IN THE PROCESS INDUSTRIES	190	General
[37]	JOURNAL OF HAZARDOUS MATERIALS	126	Accident prevention
[38]	JOURNAL OF HAZARDOUS MATERIALS	91	Chemical industry
[39]	JOURNAL OF HAZARDOUS MATERIALS	84	Accident prevention
[34]	RELIABILITY ENGINEERING and SYSTEM SAFETY	69	General
[40]	PROCESS SAFETY AND ENVIRONMENTAL PROTECTION	67	Accident prevention
[36]	ENGINEERING FAILURE ANALYSIS	61	Road
[41]	MATERIALS TESTING	55	Road
[42]	ENGINEERING FAILURE ANALYSIS	54	Road
[43]	JOURNAL OF LOSS PREVENTION IN THE PROCESS INDUSTRIES	53	Offshore oil and gas

**Table 3 ijerph-19-09512-t003:** Definitions of safety barriers.

Sources	Definition	Industry
[9]	Safety barriers are physical and/or nonphysical means to prevent, control, or mitigate undesired events or accidents.	General
[37]	Safety barriers can be physical and engineered systems or human actions based on specific procedures or administrative controls.	General
[34]	Safety barrier is defined as a series of elements dedicated to a certain barrier function where the element can be technical or human-related.	Chemical industry
[61]	Safety barrier refers to measures to protect vulnerable assets against hazards posed by failures or deviations of systems.	General
[62]	Safety barriers are defined as systems of technical, operational and organizational elements, which individually or collectively reduce the possibility for a specific error, hazard or accident to occur or which limit its harm/disadvantages.	Offshore oil and gas
[63]	The safety barrier is designed to reduce the frequency and severity of a top event.	Offshore oil and gas
[64]	Safety barriers contain components to protect, mitigate and prevent hazardous sequences of events.	Offshore oil and gas
[65]	Safety barrier refers to physical and nonphysical means implemented to reduce the possibility of technological accidents or to lessen their impact	Natech scenario
[66]	Safety barrier refers to a system describing the means by which the barrier functions are carried out.	General
[67]	The safety barrier is used to describe all aspects associated with safety, such as functions, elements and systems.	Maritime shipping

**Table 4 ijerph-19-09512-t004:** Indicators or aspects used in safety barrier performance assessment.

Source	Aspects or Indicators	Industry
[37]	Effectiveness, response time, level of confidence	General
[77]	Effectiveness, reliability, availability	Accident prevention
[64]	Effectiveness, degree of confidence, economic impact	Chemical industry
[21,76]	Availability (active barrier), effectiveness (passive barriers)	Natech scenarios
[62]	Reliability, effectiveness and robustness	Offshore oil and gas
[46]	Availability, effectiveness	Offshore oil and gas
[66]	Efficiency, resource needs, robustness, availability, independence	General
[10]	Availability, probability of failure on demand	Chemical industry

**Table 5 ijerph-19-09512-t005:** Definitions and associated operators for different gate types [49].

Gate Type	Probability Distribution	Graphical Representation
a	Simple composite probability: the PFD is multiplied by a single probability of the barrier’s success in the prevention of the domino effect.	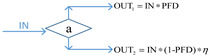
b	Composite probability distribution: the PFD is multiplied by a probability distribution expressing the probability of the barrier preventing the domino effect successfully.	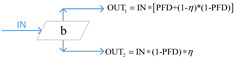
c	Discrete probability distribution: the PFD is multiplied by a discrete probability expressing the probability of the barrier preventing a domino effect in which at least three barriers are involved.	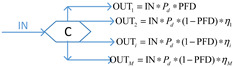
d	Fragility gate: the output is determined by the objective equipment with the application of equipment vulnerability models.	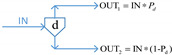

**Table 6 ijerph-19-09512-t006:** Standards and guidelines associated with barrier management.

Source	Standards or Guidelines	Industry
[62]	Principles for barrier management in the petroleum industry	Offshore oil and gas
[136]	S-001 Technical Safety (Edition 4)	Offshore oil and gas
[137]	Guidelines for barrier management in the petroleum industry	Offshore oil and gas
[114]	Guidelines for Bow-tie risk management	Chemical industry
[115]	Human Factors in Barrier Management	Generic (human-related)
[83]	Safety instrumented systems for the process industry sector	Chemical industry
[138]	Safety of machinery—Functional safety of safety-related electrical, electronic and programmable electronic control systems	Chemical industry
[139]	Road safety barrier systems and devices—Road safety barrier systems	Road transportation
[140]	Railway applications—The specification and demonstration of Reliability, Availability, Maintainability and Safety	Railway transportation
[141]	Machine Safety—Preventive Fire Protection and Protection	Generic
[142]	Petroleum and natural gas industries—Well integrity	Offshore oil and gas
[143]	Petroleum and natural gas industries—Drilling and production equipment and subsurface barrier valves and related equipment	Offshore oil and gas

## Data Availability

Not applicable.

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
