# Peer review of "Barriers Involved in the Safety Management Systems: A Systematic Review of Literature"

_ijerph, 2022, doi:10.3390/ijerph19159512_

Round 1

Reviewer 1 Report

This article reviews literature on barriers to  reduce risk. It is comprehensive and excellently structured and written. It shows a full mastering of the material. There is little to comment upon.

It is doing more than reviews usually do: it defines clear issues, it proposes a new concept - risk-based capacity, and suggests research to further improve.

Why do you want to publish this in a journal dedicated to occupational health?  From the nature of the content of the paper it is more suited for a journal on process safety. Safety barriers address of course occupational safety too, but the far majority of papers your reviewed are concerned with process safety failures.

A further comment that can be made is: why is suggestion 4.5 containing use of big data for health monitoring, not following 4.3 on Industry 4.0, so that suggestion 4.5 under the heading of resilience comes last? And to come last is the goal of resilience assessment. It should be the umbrella to guard against the unexpected and unknown, and it will comprise early warning, management alertness, effective response and recovery, so it goes beyond barriers, and it addresses the entire socio-technical system. 

Minor comment: 

It is common to write socio-technical system instead of social-technical system.

Author Response

Dear Editors and Reviewers:

Thank you for your  comments concerning our manuscript entitled “Barriers involved in the safety management systems: A systematic review of literature”. The authors are very appreciative of the valuable comments and recommendations provided for the improvement of the above-referenced manuscript. We have studied comments carefully and have made correction which we hope meet with approval. Revised portion are marked in red in the manuscript. And the responses to your comments can be found in the attached file.

Once again, thank you very much for your comments and suggestions.

Reviewer 2 Report

Barriers involved in the safety management systems: A systematic review of literature 

This paper is a review of the literature on barriers for safety in safety management systems. This is an interesting and important topic with which I am relatively familiar. The paper is well written and provides important information. However, there are some substantial issues that would need to be addressed before this would be appropriate for publication. 

1. The first issues is that there is no clear argument made in the introduction for the need for this type of review of the literature. Lines 87 - 108 discuss the aim of the paper but do not indicate at all any type of gap that currently exists and how a review (and synthesis) of the literature would facilitate filling that gap. 

2. I’m not sure why you used the criteria you list in lines 129 - 134. They may be fine but there is no indication why these are the appropriate criteria. 

3. Table 1. [note for this and all tables, they are measurably more readable if the text is left aligned and not centered]

4. Table 1. You have only 2 keywords (phrases) listed. Are these the only ones you used. Why?

5. Line 143. How can you have a hypothesis for this paper? This section is not clear at all. 

6. Figure 3. Need to spell out “NATECH” when it is first mentioned. This is an acronym and not all will know what it stands for. 

7. Figure 3. Better as a bar chart than this circle. 

8. Section 2.2. I have never seen in a literature review so much information provided about authors, their affiliations, journals, and keywords. The benefit of this content, particularly given that it is 10 pages(!!!) of the manuscript is not clear at all and honestly, it’s really distracting. Further, the affiliation graphs are very hard to understand or even read. This content could be interesting I’m guessing but would need to clarified and have the relevance clearly established. Also, for figure 6, the legend is covering part of the graph. For figure 8, it is not clear where those numbers came from. 

9. [btw, this is where the content I expect to see in a literature review begins] section 3.1 Why label this “Topic”? Ostensibly, you would be addressing your “hypotheses” here but you had three of those and two topics and of course the topics don’t correspond to your hypotheses. Was it that the literature tended to group into these two topics? If so, you should say that somewhere.  

10. Line 512, I don’t understand what you mean by “withstanding time”

11. Section 3.2.1.2 - this section would benefit from a discussion of human reliability analysis which is often used in QRA. (e.g., Ade, N., Ahmed, L., Espinosa, C., Koirala, Y., Peres, S. C. & Mannan, M. S. (2022) A review of human reliability assessment methods for proposed application in quantitative risk analysis of offshore industries. International Journal of Industrial Ergonomics)

12. Line 993 - 997, I agree with your summary but I don’t see your point. You state this as if it’s a problem and I am not clear what the problem is. 

Issue 2 is well stated

13. Section 4, do you mean Need for Future Research on Safety Barriers? This is my impression from reading this section. 

14. Your section 4 is really good and a real contribution. I think generally that the paper is so dense it’s a bit hard to get to. Most literature reviews are dense so that’s hard to avoid but I think giving the readers the map or outline of what the paper is going to cover at the beginning, much like you do at the end, would be really helpful. For instance, if the paragraphs before 2. were to be about why this review is needed and then about how the paper is organized, the reader would be much better prepared for what is to come (however, I still contend that the content of 2.2 - 2.5 below in another publication).

Author Response

Dear  Reviewers:

Thank you for your  comments concerning our manuscript entitled “Barriers involved in the safety management systems: A systematic review of literature”. The authors are very appreciative of the valuable comments and recommendations provided for the improvement of the above-referenced manuscript. We have studied comments carefully and have made correction which we hope meet with approval. Revised portion are marked in red in the manuscript. And the responses to your comments can be found in the attached file.

Once again, thank you very much for your comments and suggestions.
